# Evaluation of Mechanical and Physical Properties of Light and Heat Polymerized UDMA for DLP 3D Printer

**DOI:** 10.3390/s21103331

**Published:** 2021-05-11

**Authors:** Qutaiba Alsandi, Masaomi Ikeda, Yoshinori Arisaka, Toru Nikaido, Yumi Tsuchida, Alireza Sadr, Nobuhiko Yui, Junji Tagami

**Affiliations:** 1Cariology and Operative Dentistry, Graduate School of Medical and Dental Sciences, Tokyo Medical and Dental University (TMDU), 1-5-45 Yushima, Tokyo 113-8549, Japan; dr.alsendi@gmail.com (Q.A.); tagami.ope@tmd.ac.jp (J.T.); 2Oral Prosthetic engineering, Graduate School, Tokyo Medical and Dental University (TMDU), 1-5-45 Yushima, Tokyo 113-8549, Japan; 3Department of Organic Biomaterials, Institute of Biomaterials and Bioengineering, Tokyo Medical and Dental University (TMDU), 2-3-10 Kanda-Surugadai, Chiyoda, Tokyo 101-0062, Japan; arisaka.org@tmd.ac.jp (Y.A.); yuiorg@tmd.ac.jp (N.Y.); 4Department of Operative Dentistry, Division of Oral Functional Science and Rehabilitation, School of Dentistry, Asahi University, 1851 Hozumi, Mizuho-city, Gifu 501-0296, Japan; nikaido-ope@dent.asahi-u.ac.jp; 5Department of Restorative Dentistry, School of Dentistry, University of Washington, 1959 NE Pacific St, Seattle, WA 98195, USA; yumi.bmoe@tmd.ac.jp (Y.T.); arsadr@uw.edu (A.S.)

**Keywords:** 3D printing, digital dentistry, operative dentistry

## Abstract

The aims of this study were to investigate the feasibility of using a DLP 3D printer to fabricate a crown using scan data before tooth preparation, and to investigate the effect of additional heat curing on the mechanical properties of the urethane dimethacrylate (UDMA)-based 3D printed crown. A silicone fitting test was used to evaluate the internal adaptation of the crown. For ultimate tensile strength (UTS), the specimens were tested after 24 h storage in water at 37 °C or after 10,000 thermal cycles (TC) between 5–55 °C. For shear bond strength (SBS), a PMMA self-curing resin was filled into a Teflon ring mounted onto the polished UDMA specimens. The internal adaptation of the crowns fabricated with cement space was better than those with no cement space. There was no significant difference in UTS between light-curing and additional heat-curing groups after TC. As for the SBS, there was a significant difference after TC between the two groups. Crowns can be fabricated by a DLP 3D printer using pre-preparation scans with a cement space defined in the software. Additional heat curing of the UDMA-based crown reduced residual monomer and improved its mechanical properties.

## 1. Introduction

Digital technology has provided novel methods that are rapidly progressing, presenting new treatment options in dentistry. Computer-aided design, computer-aided manufacturing (CAD/CAM), three-dimensional (3D) printers, and intra-oral scanners are considered among the popular digital technologies in the dental field [1,2]. With digital models, dental laboratory tasks have become easier and faster than the traditional methods using impression material and plaster models, owing to the low technique sensitivity, high accuracy, and stability in digital dentistry [2,3,4]. However, the high expense and large amount of material waste in CAD/CAM using the milling method have made digital 3D printers a desirable alternative [5,6].

Three-dimensional (3D) printing or additive manufacturing is an inclusive term for several modalities. Among 3D printers, the digital light processing (DLP) printers are the most utilized method in dentistry. Models made by the stereolithography (SLA) method are more accurate than those made by DLP, however, the process of photopolymer curing in DLP printers is considered faster than SLA, since SLA uses a laser light source and DLP uses a Light Pattern Generator. This method can reach 60 μm of accuracy according to some manufacturer data [6,7]. In addition, DLP printers have a digital projector screen used to transfer a single image of each layer across the entire build plate at once [6,8,9].

These printers use curable photopolymer resin to fabricate the models. Since the option of using a monomer is important for the formulation of 3D printing resins for dental use, choosing monomers commonly found in light-polymerized dental resins, including bisphenol A-glycidyl methacrylate (Bis-GMA) and urethane dimethacrylate (UDMA), is a reasonable approach [8,10,11]. However, the effectiveness of the light polymerization process in terms of establishing the mechanical properties and the amount of residual monomers after the process has always been a concern for these materials.

Currently, there are various methods of creating provisional crowns. Many practitioners still use the traditional direct or indirect fabrication technique using self-polymerizing acrylic resins and indexes created using waxed-up models or pre-preparation impressions. More recently, 3D printers have been used to print the model (s) that substitute the traditional plaster model, but these methods are still time-consuming. Fabrication of a provisional crown directly using the 3D printer with no or minimal need for the crown design is ideal. However, there are few reports describing such a technique that would use the 3D scan data prior to tooth preparation to fabricate a 3D printed provisional crown.

The aim of this study was to investigate the feasibility of fabricating the provisional crown using pre-preparation and preparation 3D data. Furthermore, the second aim of this study was to investigate if the mechanical and physical properties of UDMA-based dental resin would be affected by additional heat curing.

## 2. Materials and Methods

### 2.1. Materials Used in This Study

Table 1 lists the materials used in this study, their composition, manufacturers, and fabrication methods. A digital light processing (DLP) type 3D printer (Zenith D, Dentis, CA, USA) was used to fabricate the specimens. A light-curing resin material (Zenith Temporary, Dentis, CA, USA) was used in the DLP, which is composed of urethane dimethacrylate (UDMA). Besides that, a conventional dental self-curing resin (Unifast III, GC, Tokyo, Japan; PMMA) was used, which is composed of the monomer (methyl methacrylate; MMA), the polymer (poly methyl methacrylate; PMMA), and the catalyst (Barbituric acid derivative, Dilauryl-dimethyl-ammonium chloride, and Acetylacetone copper).

### 2.2. Specimens Preparation

A DLP 3D printer was used to fabricate the specimens using the UDMA material. A CAD/CAM software (Zenith D Slicer, Dentis, CA, USA) was also used as a controller software for the 3D printer to modify printer settings such as support patterns, layer thickness, and layer angle of the specimens. STL data of the shear bond and ultimate tensile test specimens were prepared accordingly using 3D design and modeling software (Geomagic Freeform modeling Plus, 3D Systems, Rock Hill, SC, USA).

### 2.3. Fabrication of Crown

Figure 1 shows the fabrication of a crown. For the provisional crown fabrication, an intraoral scanner (Torios 3, 3 shape, Copenhagen, Denmark) was used to obtain the 3D data of the unprepared lower right first molar tooth model (A5A-500, Nissin, Kyoto, Japan) and a prepared lower right first molar tooth model (A55A-461, Nissin, Kyoto, Japan). After that, 3D data of the unprepared and prepared teeth were superimposed by the CAD software (Geomagic Freeform modeling Plus, 3D Systems, Rock Hill, SC, USA), the 3D data of the prepared tooth were subtracted from those of the unprepared tooth using the same CAD software. Two groups were created using the created crown data: with and without a cement space (CS and non-CS). In the CS group, the cement space of the occlusal surface was set at 10% of the crown thickness and for the mesial, distal, buccal, and lingual surface it was set at 5% using CAD software (3D builder, Microsoft, WA, USA).

Then, the STL files generated from the designing process were transferred into the CAM software (Zenith D Slicer, Dentis, CA, USA). The printing parameters were set as follows: support patterns (8 cone structures on the occlusal surface), layer thickness (100 µm), and layer angle (0°) for the fabrication of crowns. Then, the provisional crown was printed using a DLP 3D printer (Zenith D, Dentis, CA, USA) with a light-curing resin (Zenith Temporary, Dentis, CA, USA). The 3D printer cured the resin at under 40 °C temperature. Later, all the specimens were divided into two groups according to the curing method: light cured by a light-curing unit (Zenith Cure, Dentis, CA, USA) and light cured with additional heat curing at 110 °C for 15 min by a heat-curing unit (Pearl cure heat, Tokuyama dental, Tokyo, Japan). In total, 240 crown specimens were fabricated for the internal adaptation test.

### 2.4. Evaluation of Internal Fit of the Crown

An internal-fitting test was carried out for the evaluation of the internal fit of the crown on the prepared tooth model [12]. Silicone rubber impression materials (Fit Checker Base and Catalyst, GC, Tokyo, Japan) were mixed, and the white-colored mixture was poured into the inner surface of the crown. The crown was then rapidly fitted onto the prepared tooth model (A55A-461, Nissin, Kyoto, Japan). Finger pressure was applied on the occlusal surface of the crown for 3 min before the complete setting of the impression material. Afterward, the crown and impression material were carefully removed from the prepared tooth model. A different set of silicone rubber impression materials (Bite Checker Base and Catalyst, GC, Tokyo, Japan) was then mixed, and the resulting black-colored mixture was poured into the inner surface of the crown, over the previously placed Fit Checker. After 3 min, the white (Fit Checker) and the black (Bite Checker) silicone blocks were carefully removed from the crown. The silicone blocks were divided in half at the middle using a stainless-steel razor blade (Feather, Osaka, Japan) for cross-section observation [13].

The cross-sectional area was imaged under 4.0× magnification by using an optical microscope (LM, Nikon SMZ1000, Nikon Corp., Tokyo, Japan). Image data were loaded to the image analysis software (ImageJ, version 1.48, National Institution of Health, Bethesda, MD, USA). For the evaluation of internal fit, five standard measurement points were set (P0, P1, P2, P3, and P4) on the surface of the cross-sectional area.

The thickness of the white silicone material was measured twice at the five standard measurement points: (P0) the margin, (P1) 150 µm from the margin, (P2) bisection point between P1 and inflection point, (P3) bisection point between inflection and the highest point, and (P4) bisection point between the highest and the central pit [13] (Figure 2) using the image analysis software (ImageJ, version 1.48, National Institution of Health, Bethesda, MD, USA). Then the mean thickness was calculated in each point (P0, P1, P2, P3, and P4) of the buccal and lingual surface.

### 2.5. Ultimate Tensile Strength (UTS) Test

A total of 52 dog-bone-shaped samples (Figure 3) were fabricated according to ISO 527-2-1BB specifications using the DLP 3D printer. The printing parameters were set as follows: support patterns (10 cone structures on the surface of the specimen), layer thickness (100µm), and layer angle (0°) for the fabrication of UTS specimens. All the specimens were divided into two groups according to the curing conditions: light cured by a light-curing unit (Zenith Cure, Dentis, CA, USA) and light + heat cured at 110 °C for 15 min by a heat-curing unit (Pearl cure heat, Tokuyama dental, Tokyo, Japan). Those groups were also divided into subgroups according to the storage conditions: one group was stored for 24 h in distilled water at 37 °C and the next group was subjected to thermal cycling for 10,000 TC (5 – 55 °C, dwell time for 30 s each). Then, all the specimens were subjected to loads to measure the UTS using a universal testing machine (Autograph AGS-J, Shimadzu, Kyoto, Japan) at a crosshead speed of 5 mm/min.

### 2.6. Shear Bond Strength (SBS) Test

A total of 52 disc-shaped specimens (d= 20 mm, h = 10 mm) were made for SBS. The printing parameters were set as follows: support patterns (12 cone structures on the bottom surface of the specimen), layer thickness (100 µm), and layer angle (0°) for the fabrication of SBS specimens. All the specimens were divided into two groups according to the conditions of curing: light cured and light + heat cured as described above. The surfaces of all the specimens were ground flat using #600-grit SiC paper under running water and embedded into the self-curing resin PMMA. A ring (inner = 3 mm, h = 4 mm) of polytetrafluoroethylene (Teflon) was then mounted to determine the adhesive area. The self-cured acrylic resin powder and liquid were mixed according to the manufacturer’s instructions and poured into the Teflon^@^ tube at room temperature. Then, the Teflon^@^ tube was carefully removed after the complete setting of the acrylic resin. After preparing all the specimens, the light-cured group and light + heat cured group were split into subgroups according to the storage conditions as described for the UTS specimens. Finally, the shear bond strength test specimen was loaded at a crosshead speed of 1 mm/min using a universal testing machine (Autograph AGS-J, Shimadzu, Kyoto, Japan).

### 2.7. FT-IR Analysis of Unpolymerized Monomers

The residual percentage of unpolymerized UDMA on the surfaces of the shear bond strength specimens was determined by a Fourier transform infrared (FT-IR) spectrometer (Spectrum 100, Perkin Elmer, Waltham, MA, USA). The resin material (Zenith Temporary, Dentis, CA, USA) was prepared using the same specimen preparation method as SBS. An additional heat-cured group was created using 55 °C for this experiment. All the specimens’ surfaces were ground using a carbide bur (14NF, Shofu, Kyoto, Japan) to obtain powder specimens. Each sample was ground and blended with KBr powder to prepare pellets for FI-IR measurements. All the spectra were recorded in the frequency range of 700–4000 cm^−1^ at a resolution of 4 cm^−1^ and analyzed using Spectrum software (PerkinElmer). In order to evaluate the residual percentage of UDMA in specimens, an absorption peak of the carbon–carbon double bonds (C=C) at 1638 cm^−1^ was compared with the absorption of amide II derived at 1532 cm^−1^ [14]. The peak intensity ratio (I_1638_/I_1532_) of UDMA (436,909-100 mL, Sigma Aldrich, MO, USA) was used as a control for calculating the residual percentage.

### 2.8. Statistical Analysis

The distribution and variance of data were analyzed using the Shapiro-Wilk test and the Levene test. UTS and SBS data were analyzed by two-way ANOVA and t-test with Bonferroni correction, as data showed both normal distribution and equal variances. The data on curing conditions and cement space were analyzed by the Wilcoxon rank sum test. Furthermore, the data on the five standard measurement points were analyzed by Wilcoxon signed rank test with Bonferroni correction, since the data appeared to have a normal distribution. The significance level was set at alpha = 0.05. All statistical procedures were performed using Statistical software (SPSS ver. 22.0 for Windows, IBM Corp., Chicago, IL, USA).

## 3. Results

### 3.1. UTS Measurement

The results of the UTS test are summarized in Table 2. The UTS after 10,000 TC was significantly higher than that after 24 h in the light-cured group (*p* < 0.05). However, the UTS after 10,000 TC was significantly lower than that after 24 h in the light- + heat-cured group (L + H) (*p* < 0.05). There was a significant difference between the light-cured group (L) and the light- + heat-cured group (L + H) after 24 h distilled water storage (*p* < 0.05), but no significant difference after 10,000 TC (*p* > 0.05).

### 3.2. SBS Measurement

The results of the SBS test are shown in Table 3. There was no significant difference between the light-cured group (L) and the light- + heat-cured group (L + H) after 24 h (*p* > 0.05). However, the SBS significantly increased after 10,000 TC in the light-cure group (*p* < 0.05), while the SBS significantly decreased after 10,000 TC in light- + heat-cured group (L + H) (*p* < 0.05).

### 3.3. Internal Adaptation Test

Table 4 shows the result of the internal adaptation test for the five reference points for the crown (see Figure 2). For P0, there were significant differences in the thickness of the internal gap among the four groups (*p* < 0.05). The thickness of the internal gap was higher in the non-CS group than in the CS group (*p* < 0.05). The thickest value was obtained in the non-CS group with light-curing mode (L) (82 µm). For P1, there was no statistically significant difference among the groups (*p* > 0.05). On the other hand, similar tendencies were observed in P2 and P3, in which there were no significant differences between the light-curing mode (L) and the light- and heat-curing mode (L + H) in the non-CS and CS groups, however, the internal gap was higher in CS than in non-CS (*p* < 0.05). For P4, there were significant differences in the thickness of the internal gap among the four groups (*p* < 0.05). The thickness of the internal gap was higher in the non-CS group than in the CS group (*p* < 0.05) in both the light-curing mode (L) and the light- and heat-curing mode (L + H). The thickest value was obtained in the non-CS group with light-curing (L) (71 µm).

Regional differences for each reference point were observed in the non-CS groups, but not in the CS groups. A thicker internal gap was measured at P0 and P4 than at P1, P2, and P3 in the non-CS group with both the light-curing mode (L) and the light- and heat-curing mode (L + H) (*p* < 0.05).

### 3.4. FT-IR Spectra

Figure 4 shows the typical FT-IR spectra of the specimen surfaces. The monomers of the UDMA-based material (Zenith Temporary, Dentis, CA, USA) were used without polymerization as a positive control group (Unpolymerized monomer 100%), while the specimens after light and heat curing at 110 °C were used as the negative control group (Light + heat cure, 110 °C). From the calculation of the intensity ratio of the absorption peak (I_1638_/I_1532_), the percentages of the residual UDMA monomers in the light-cure mode and light- and 55 °C heat-cure mode were 40.4% and 8.8%, respectively.

## 4. Discussion

Digital light processing (DLP) is one of the most favorable 3D printing technologies for dental purposes because of its fast processing speed, high resolution, and low cost of the printer and its materials (Lin et al., 2019; Revilla-León and Özcan, 2019). The light coming from a DLP projector during printing supplies energy to polymerize photosensitive materials layer by layer (dos Santos et al., 2008; Lin et al., 2019; Ruyter and Øysæd, 1982). UDMA and other methacrylate monomers are used in those kinds of 3D printers (Al Mousawi et al., 2018a, 2018b; Bagheri and Jin, 2019; Yue et al., 2015). The UDMA-based material contained a photo-polymerization initiator and no heat-polymerization initiator, and heat polymerization after light polymerization of UDMA-based material to improve physical properties has not been reported.

The UTS value of the light- + heat-cured group was significantly higher than that of the light-cured group after 24 h. This can be explained by the improvement of the polymerization process by heat curing at 110 ℃. However, the UTS value of the light- + heat-cured group significantly decreased after 10,000 TC. The thermocycling test is known to accelerate the hydrolytic degradation of polymer-based materials by water penetration and heat stress at 5 –55 ℃ [14]. On the other hand, the UTS value of the light-cured group significantly increased after 10,000 TC, which may be explained by the improvement of the polymerization of the specimens during exposure to heat at 55 ℃.

The PMMA self-curing resin is used for provisional restoration in clinics. The gingival discrepancy, proximal contact, and occlusion contact are modified by self-curing resin. Therefore, it is necessary to evaluate adhesion by SBS between UDMA and PMMA. From the SBS results of this study, there was no significant difference between the light-curing group and the light- + heat-curing group after 24 h. However, the tendency of the SBS values was quite different between the two groups. For the light-curing group, the SBS value significantly increased after 10,000 TC, while the SBS value for the light + heat cure decreased after 10,000 TC. The different tendency may be due to the residual monomer on the surface of each specimen. The results of FT-IR indicated that the unpolymerized UDMA monomers and the double bond (C=C) that remained on the specimen surface were different between the light-cured group and the light- and heat-cured group. Considering the bonding performance of the self-cured acrylic resin to the UDMA-based specimen, the residual monomer of the UDMA-based specimens is beneficial for the self-cure acrylic resin to bond to the specimen surface. Adhesion of the self-cured acrylic resin to the unpolymerized layer on the light-curing resin material may be due to an interpenetrating polymer network [15,16,17]. The interpenetrating polymer network at the adhesive interface must be enhanced by heat curing at 55 ℃ temperature in water during the thermocycling test. On the other hand, the results of FT-IR indicated that little unpolymerized UDMA monomers remained on the surface of the specimen after heat curing at 110 ℃ for 15 min. Therefore, the SBS of the light- + heat-curing group was significantly decreased after 10,000 TC.

The crown shape before the preparation was scanned by an intra-oral scanner. This technique has advantages over the design of the crown using the 3D software, in terms of occlusion, anatomy, and proximal contacts, and will eventually save time for the fabrication of the crown. However, the pre-preparation anatomy may not always be readily suitable for use for fabrication of the provisional, in such cases as temporary build up and fractured or worn-out tooth, and 3D manipulation of the pre-preparation data may be necessary to create the desired anatomy for the provisional crown. Furthermore, the mechanical properties of 3D printed specimens are influenced by support patterns, layer thickness, and layer angle [18]. The number of support structures, structure shape, layer thickness (50 or 100 µm), and layer angle (0–90°) can be changed in Zenith D CAM software. In this study, eight cone structures on the occlusal surface, a layer thickness of 100 µm, and a layer angle of 0° were used for the fabrication of crown specimens. These were decided based on the performance in terms of the fabrication speed of crowns in a clinic.

The first aim of this study was to fabricate the crown using 3D data from before and after the preparation of the tooth. The results suggested that the crown fit was not great for the on-abutment tooth using tooth data without cement space. This was attributed to the deformation of the inner surface caused by polymerization shrinkage of the 3D printed resin in this study. It was reported that UDMA polymerization resulted in shrinkage by light or heat curing [19,20]. A plaster (stone) model is usually used to fabricate provisional restorations [21,22]. In the conventional method, deformation by curing shrinkage may be physically limited to the inner surface of the crown. The results of light curing with the cement space group showed better adaptation than light curing in the without cement space group. The results indicate that the curing shrinkage was compensated due to cement space. Without cement space, there was a better adaptation when additional heat curing was applied. However, there was no significant difference between the light-curing group and the light- + heat-curing group with cement space.

In this study, additional heat curing was found to influence the results of UTS, SBS, and internal adaptation. In particular, the curing process by thermocycling at 55 ℃ tended to increase UTS and SBS of only light-cured specimens, because the UDMA material using a 3D printer was not completely cured by the light-curing unit. The provisional material used in this study was evaluated for biocompatibility based on ISO 10,993 (Resin for Temporary crown & Bridge, Premarket Notification: Traditional 510(k) Number K180675). However, in the result of the FT-IR analysis, 40% uncured monomer remained on the surface of the specimen. It is preferable that a reactive substance is not contained in 3D printing materials for use in the human body [23,24]. Therefore, the uncured component of provisional material should be reduced as much as possible. Based on these results and within the limitations of the present study, it may be concluded that heat treatment does not improve the UTS and SBS. However, heat treatment improved internal adaptation and the degree of conversion of UDMA material on the surface of specimens. So, it is plausible to suggest that the combination of light curing and heat curing may be an alternative to achieve better results, although further studies are needed to verify this application in daily clinical practice.

## 5. Conclusions

This study defined a cement-space-improved internal adaptation of provisional crowns fabricated using pre-preparation 3D scans. The technological development is expected to automatically calculate by CAM software the cement space according to the shape of the abutment tooth.

The UTS showed no significant difference between L and L + H after thermal cycling. The SBS significantly decreased after thermal cycling. The uncured component of provisional material should be reduced as much as possible with additional heat curing.

## Figures and Tables

**Figure 1 sensors-21-03331-f001:**
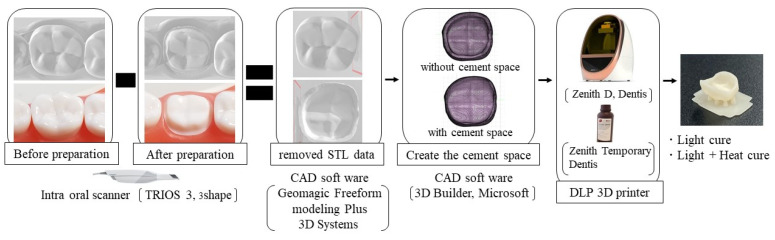
Fabrication of crown.

**Figure 2 sensors-21-03331-f002:**
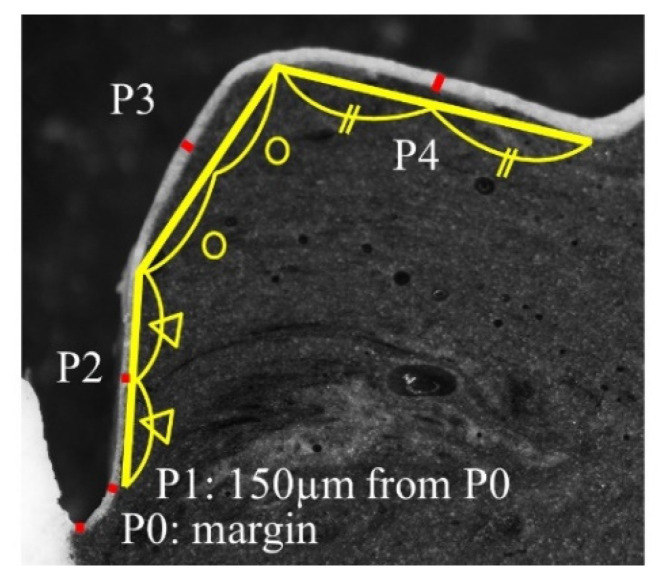
Reference points for evaluation of internal adaptation of the crown.

**Figure 3 sensors-21-03331-f003:**
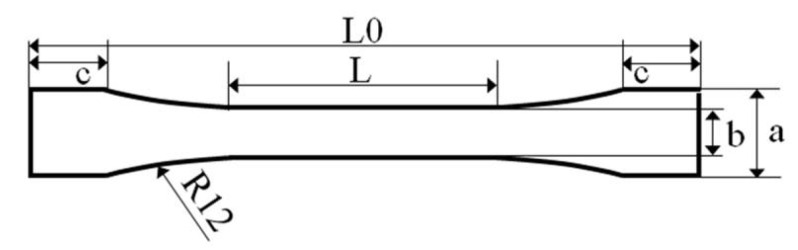
Ultimate tensile test specimen. (mm) a: 4.0 ± 0.1, b: 2.0 ± 0.1, c: 3.5 ± 0.1, thickness: 2.0 ± 0.2, L: length of reduced section 12.0 ± 0.2, L0: overall length 30.0 ± 0.2

**Figure 4 sensors-21-03331-f004:**
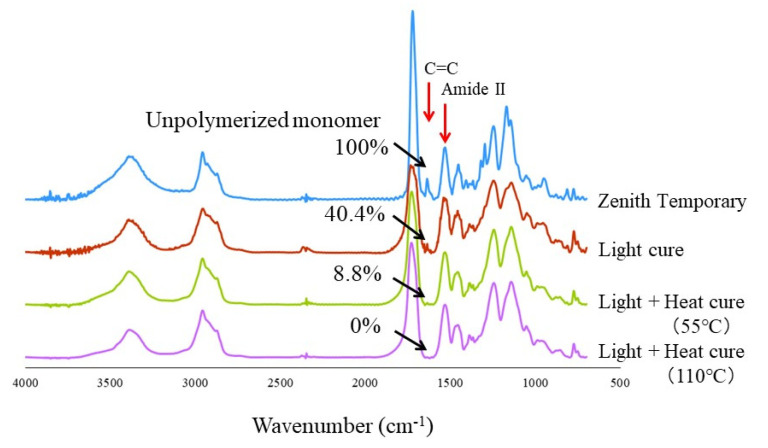
FT-IR spectra of the specimen surfaces.

**Table 1 sensors-21-03331-t001:** Devices and materials.

Devices, Software, and Materials	Properties	Manufacturer
**Intraoral Scanner (Trios 3)**	Open-data for STL	3shape (Denmark)
**3D Printer (Zenith D)**	Digital light processing 3Dprinter LED light source: 405 nm	Dentis (USA)
**Light-curing resin** **(Zenith Temporary)**	UDMA, Methacrylate oligomer based on the Urethane Acrylate Oligomer 0.01~0.1 wt% inorganic filler Photo-polymerization initiator	
**Zenith D CAM software** **(Zenith D Slicer)**	Controller software for the 3D printer Manipulate printing settings such as: support patterns, layer thickness, layer angle	
**UDMA** **(436909-100 mL)**	Diurethane dimethacrylate, mixture of isomers contains 225 ppm ± 25 ppm topanol as inhibitor, ≧97%, Lot No. MKCG8230	Sigma-Aldrich (USA)
**3D design and modeling software** **(Geomagic Freeform modeling Plus)**	Subtracting the 3D data of prepared tooth from the 3D data of the unprepared tooth	3D Systems (USA)
**3D design and modeling software** **(3D Builder)**	Creating the cement space on the inner surface of the crown	Microsoft (USA)
**Self-cured acrylic resin** **(Unifast III)**	Powder: Methyl Methacrylate–Ethyl Methacrylate copolymer, Barbituric acid derivative, and Acetylacetone copper, Benzoyl peroxide Liquid: Methyl Methacrylate monomer, Dilauryl-dimethyl-ammonium chloride and Hydroquinone	GC (Japan)

**Table 2 sensors-21-03331-t002:** Ultimate tensile strength of the UDMA based material used in this study.

Curing Conditions	24 h	TC10,000
L	31.72 ± 3.70 ^a,A^	34.75 ± 1.69 ^a^
L + H	40.49 ± 2.78 ^b,A^	34.31 ± 2.23 ^b^

L: Light cure, L + H: Light cure and heat cure at 110 °C for 15 min. 24 h: 24 h in distilled water at 37 °C, TC: 10,000-Thermal cycling (5–55 °C, dwell time for 30 s each) after 24 h in distilled water at 37 °C. Same small and large script letters indicate significant differences in horizontal rows and vertical columns (*p* < 0.05).

**Table 3 sensors-21-03331-t003:** Sear bond strength of the UDMA based material and PMMA used in this study.

Curing Conditions	24 h	TC10,000
L	14.51 ± 2.37 ^a^	20.96 ± 3.77 ^a,A^
L + H	15.69 ± 4.22 ^b^	12.35 ± 5.32 ^b,A^

L: Light cure, L + H: Light cure and heat cure at 110 °C for 15 min. 24 h: 24 h in distilled water at 37 °C, TC: 10000-Thermal cycling (5 –55 °C, dwell time for 30 s each) after 24 h in distilled water at 37 °C. Same small and large script letters indicate significant differences in horizontal rows and vertical columns (*p* < 0.05).

**Table 4 sensors-21-03331-t004:** Internal adaptation for five reference points were chosen for crowns.

Curing Conditions	Cement Space	P0	P1	P2	P3	P4
L	Non-CS	81.88 ± 14.92 ^a,b,c,A,C^	47.79 ± 14.64 ^a^	28.84 ± 5.26 ^b,d,D^	38.88 ± 5.68 ^c,F^	70.59 ± 27.73 ^d,H,I^
	CS	35.73 ± 8.01 ^h,I,A^	37.62 ± 5.59 ^j^	45.79 ± 4.72 ^h,D^	56.93 ± 7.41 ^i,j,k,F^	40.96 ± 7.27 ^k,H^
L + H	Non-CS	59.22 ± 16.57 ^e,f,g,B,C^	34.25 ± 11.51 ^e^	28.48 ± 7.88 ^f,E^	33.22 ± 8.78 ^g,G^	44.51 ± 14.24 ^I^
	CS	38.88 ± 5.99 ^l,B^	37.26 ± 6.83 ^m^	42.44 ± 3.83 ^l,E^	53.37 ± 6.19 ^m,n,G^	37.93 ± 6.13 ^n^

L: Light cure, L + H: Light cure and heat cure at 110 °C for 15 min. CS: Cement space of occlusal inner surface was created at 10% of crown thickness and the medial, distal, buccal, and lingual surfaces were created at 5%. (P0) the margin, (P1) 150 µm from the margin, (P2) bisection point between P1 and inflection point, (P3) bisection point between inflection and the highest point, and (P4) bisection point between the highest and the central pit. Same small and large script letters indicate significant differences in horizontal rows and vertical columns (*p* < 0.05).

## Data Availability

Not applicable.

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
