# Peer review of "Evaluation of Mechanical and Physical Properties of Light and Heat Polymerized UDMA for DLP 3D Printer"

_sensors, 2021, doi:10.3390/s21103331_

Round 1
Reviewer 1 Report
General comment:
This manuscript investigates the feasibility of using DLP 3D printer to fabricate crown. Various characterization has been done to show the functionality of the crown. However, the reviewer thinks that the manuscript can be improved if the following comments is addressed.
Specific comments:
- The conclusion is too short. Suggest providing some insights and perspective on the future prospect of this work.
- Missing scalebar for figure 2.
- Why did the author do tensile test? why not compression test since crown will experience mostly compression force when in use.
- Formatting error in table 3.
- Since this manuscript is about 3D printing of parts used on human body, suggest citing the following manuscript:
- Bergmann, C., Lindner, M., Zhang, W., Koczur, K., Kirsten, A., Telle, R., & Fischer, H. (2010). 3D printing of bone substitute implants using calcium phosphate and bioactive glasses. Journal of the European Ceramic Society, 30(12), 2563-2567.
- Luis, E., Pan, H. M., Sing, S. L., Bastola, A. K., Goh, G. D., Goh, G. L., ... & Yeong, W. Y. (2019). Silicone 3D printing: process optimization, product biocompatibility, and reliability of silicone meniscus implants. 3D Printing and Additive Manufacturing, 6(6), 319-332.
Reviewer 2 Report
This manuscript reported the fabrication of provisional crowns using DLP 3D printing and investigation of the effect of additional heat curing on the mechanical properties of the printed crowns.
In the reviewer’s opinion, this work has not generated much new knowledge, but it may be useful for the community of dentistry and clinical research. Furthermore, it does not seem to fit to the scope of the Sensors. It may be better suited for some journals with readers in dentistry and clinical application. The following questions and comments are provided for the authors to consider.
What is the purpose of shear bond strength test? Is the printed crown bonded to other surfaces (e.g., teeth)? For the light + heat cured samples, the SBS strength significantly decreased after thermal cycling. Does it still meet the requirement of provisional crown?
Does the uncured component in the light cured samples (40.4%) pose harmful effect if it is used by patients? Any regulations associated with the extractables and leachables in the dental devices?
The authors may consider to combine the results and discussion.
There are some typos and grammar mistakes in the manuscript.
Reviewer 3 Report
The article “Evaluation of mechanical and physical properties of light and heat polymerized UDMA for DLP 3D printer” in the present form is moderately suitable for Sensors. The UDMA-based temporary crown, achieved by 3D printing, was tested for ultimate tensile strength, shear bond strength, internal adaptation, degree of conversion. These properties were tested as a function of additional heat curing. The authors have found that heat curing improves these parameters of the tested crown example.
The article can be accepted for publication after major revision.
- Grammatical and even more numerous spelling errors should be corrected.
- The aim of this study was to investigate the feasibility of fabricating the provisional crown using pre-preparation and preparation 3D data (lines 74-75). Why any information about processing parameters have not been provided? All parameters mentioned in line 94 (“support patterns, layer thickness and layer angle of the specimens”) should be specified and discussed in the results and discussion sections.
- Furthermore, the authors assumed that the mechanical and physical properties of the DLP 3D printed resin were not affected by additional heat curing (lines 76-78). Why? It is a well-known relationship that heat curing increases the degree of conversion, which rules the physicochemical and mechanical properties of the material.
- Table 1 specifies UDMA monomer from Aldrich – it is not clear how it was used.
- The Zenith Temporary composition is poorly described. It also requires citation for the datasheet.
- The application of the amide II vibration band in the FTIR spectrum as an internal standard is not common. I have not met such an approach in urethane-dimethacrylate system. It requires citation, proving that the band intensity does not change due to polymerization.
- What was the purpose of using ATR-FTIR (line 246) as KBr pellets were prepared?
Round 2
Reviewer 2 Report
The reviewer has no other comments or questions.
Reviewer 3 Report
-